# Graph Neural Network Expressivity and Meta-Learning for Molecular Property Regression

**Haitz Sáez de Ocáriz Borde**[*]
University of Cambridge

**Federico Barbero**[*]
University of Cambridge

## Abstract

We demonstrate the applicability of model-agnostic algorithms for meta-learning, specifically Reptile, to GNN models in molecular regression tasks. Using meta-learning we are able to learn new chemical prediction tasks with only a few model updates, as compared to using randomly initialized GNNs which require learning each regression task from scratch. We experimentally show that GNN layer expressivity is correlated to improved meta-learning. Additionally, we also experiment with GNN ensembles which yield best performance and rapid convergence for k-shot learning.

## 1   Introduction

Graph Neural Networks (GNNs) have recently gained attention in the machine learning community. They have achieved state-of-the-art performance in a number of tasks by leveraging the geometric prior inherent to many real-world problems [1]. Concurrently, several model-agnostic algorithms for meta-learning have been developed, such as Model-Agnostic Meta-Learning (MAML) [2] and Reptile [3]. Although as their name suggests these algorithms are *model agnostic*, works in the literature have mainly applied them to classical fully-connected and convolutional neural networks. In this paper, we explore the application of Reptile to GNN regression tasks. We show that model-agnostic algorithms for meta-learning are also applicable to GNNs and specifically, that meta-learning can exploit the underlying structure of molecules to quickly adapt models to learning new molecular regression tasks. We experimentally demonstrate that GNN expressivity is correlated to meta-learning performance. Finally, we also show that using GNN ensembles can even further improve meta-learning.

## 2   Background

Meta-learning, which can be conceptualized as *learning to learn*, enables parameter learning such that sensible predictions can quickly be elicited on new tasks from few examples [2]. This ability to perform well in data-impoverished regimes is not only reminiscent of the remarkable ability of humans to rapidly learn new concepts from limited examples [4, 5], but is especially important for applications in settings where data acquisition can be extremely costly such as healthcare [6–8], drug discovery [9, 10], robotics [11, 12], and low resource languages [13, 14]. While a diverse array of meta-learning approaches have been proposed [15, 16] such as MAML [2] and MAML++ [17], in this work, we focus on Reptile [3] for GNNs and study the effect of GNN expressivity on meta-learning. Reptile avoids some of the limitations of the original MAML algorithm, namely the computational overhead and instability issues of the MAML training procedure [3].

### 2.1   The MAML and Reptile algorithms

We first provide a primer on the methodological underpinnings of MAML [2] and build on to Reptile [3]. Following the original MAML paper [2], we consider a distribution over tasks $p(T)$, where we learn tasks $T_i$ drawn from this distribution through $K$ observations sampled from $T_i$. We

---

*Currently affiliated with the University of Oxford.*

refer to the samples used to learn task-specific parameters as the *support set*, and the samples used to evaluate such parameters as the *query set* [17]. We follow standard meta-learning terminology [2, 3, 17] in referring to evaluating generalization performance for a new task as k-shot learning, where $k$ gradient steps are taken to fit the provided observations. Moreover, we define $\alpha$ as our task-specific learning rate and $\beta$ as our meta-learning rate. MAML [2] iteratively adapts an initial set of model parameters $\theta$ based on the performance of a task-specific set of parameters $\theta'$ over a batch of tasks $T$. Specifically, for a single epoch of training, the initialization parameters $\theta$ are copied for each sampled task, $T_i \in T$. Then points are sampled in parallel from the support set per task, over which task-specific parameters $\theta'_i$ are computed. The task-specific parameter update is $\theta'_i \leftarrow \theta - \alpha \nabla_\theta L_{T_i}(f_\theta)$. Using these task-specific parameters, the yielded model is evaluated over points sampled from the query set for that task. Losses are then calculated for each individual task and pooled together. Such information, incorporating second-order gradients, is then backpropagated through the model to update the initialization parameters, via the meta-update $\theta \leftarrow \theta - \beta \nabla_\theta \sum_{T_i \sim p(T)} L_{T_i}(f_{\theta'_i})$. Note that combining both equations requires applying $\nabla_\theta$ twice, and hence second-order gradients are used to update the model parameters. For further clarification regarding the contribution of the second-order gradients please refer to [2].

Reptile [3] adopts a similar approach by attempting to identify a suitable initialization of a network. The algorithm is remarkably simple and avoids the computational and algorithmic complexity of directly dealing with second-order derivatives, bearing some of the hallmarks of FOMAML [2], while still being able to recover higher order information [3]. Reptile works by iteratively sampling a new task $T_i$ from the task distribution $p(T)$, running $k$ steps of SGD to derive new model parameters $\theta'$, and updating the initial model parameters $\theta$ using the following update equation $\theta \leftarrow \theta + \beta(\theta' - \theta)$. The authors proved that the Reptile update maximizes the inner product between gradients of different minibatches from the same task, which improves generalization and indirectly considers second-order terms [3].

## 2.2 Graph Neural Networks and Expressivity

GNNs are a class of deep learning models that operate on graph data. They leverage the additional information provided by the graph connectivity to improve inference. A GNN layer updates the latent features based on the adjacency matrix and the previous layer's node features $\mathbf{H}^{(l)} = f(\mathbf{H}^{(l-1)}, \mathbf{A})$. The message passing operation applied by many GNN layers iteratively updates node features $h_i^l \in \mathbb{R}^d$ from layer $l$ to layer $l+1$ with edge attribute information $e_{ij}$ via the following equation:

$$\mathbf{h}_i^{(l)} = \phi\left(\mathbf{h}_i^{(l-1)}, \bigoplus_{j \in \mathcal{N}_i} \psi(\mathbf{h}_i^{(l-1)}, \mathbf{h}_j^{(l-1)}, e_{ij})\right)$$

where $\mathcal{N}_i$ refers to the neighborhood of node $i$, $\bigoplus$ is a permutation-invariant aggregation function such as $\sum$ or $\max$, and $\psi$ and $\phi$ correspond to two non-linear functions which in practice can be Multi-Layer Perceptrons (MLPs).

In this work, we apply meta-learning to message passing GNN models of varying expressivity. In particular we work with convolutional, attentional, and message passing GNNs. These three *flavours* of GNNs [1] form progressively more expressive families of GNNs such that convolutional $\subset$ attentional $\subset$ message-passing, with message passing being the most expressive of all, and convolutional the least. Convolutional models use the same weighting for the neighborhood of a given node, attentional models on the other hand use different learnable coefficients for each neighbor, and message passing use a non-linear mapping to combine the features of the different node pairs. See Appendix A for more details on expressivity.

## 3 Related Work on Meta-Learning and Graph Neural Networks

Some recent works combining GNNs and meta-learning have focused on learning node and edge level shared representations [18–20]. Other contributions to the literature have concentrated on learning graph level representations instead [21, 22]. Multi-task settings involving graph classification, node classification, and link prediction using GNNs and meta-learning have also been explored [23]. The work by Guo et al [24] is particularly relevant to the topic discussed in this paper. In [24], the authors study few-shot graph learning for molecular property prediction where the tasks involve binary label

classification using the Tox21 and Sider datasets. In our case, instead of predicting binary tasks for molecules as in [24], we meta-learn quantum properties for the QM9 and Alchemy datasets. Note that none of the previous studies combine Reptile with GNNs and they do not focus on regression. Most of the existing literature adopts the MAML algorithm or derivatives to train GNNs.

Other applications combining GNNs and meta-learning include anomaly detection [25], network alignment [26], and traffic prediction [27]. Moreover, the meta-learning framework has also been used for improving the level of explainability of GNNs [28], and meta-gradients have been leveraged for adversarial attacks on GNNs [29]. For an extensive survey on meta-learning with GNNs see [30].

## 4 Experiments

We expect expressivity to be beneficial when trying to learn a model that can quickly adapt to different tasks. As message passing is the most generic and flexible GNN variety [31], we anticipate it to perform best. In this work we will focus on two related datasets. The Alchemy dataset [32] contains approximately 200,000 organic molecules and 12 quantum mechanical regression tasks. It includes molecules with a higher number of heavy atoms (C,O,N, and F) than other molecular datasets such as QM7 [33, 34], QM7b [35], QM8 [36], and QM9. We also use QM9. QM9 contains approximately 130,000 small organic molecules that may be composed of up to 9 heavy atoms. The regression targets are 19 calculated physical and chemical properties including the *Dipole moment*, and *Isotropic Polarizability*, amongst others. These datasets are chosen because they provide different regression tasks as labels. For meta-learning we train on all but one regression task, and k-shot learn to try to predict the remaining quantum mechanical property value. For both datasets, the different regression target values differ greatly in their magnitudes which can affect meta-learning performance. Hence, we normalized the regression output labels by conducting Z-score normalization [37] using the mean and standard deviation derived based on all the dataset regression targets (further details are provided in Appendix C).

### 4.1 Model Architectures

We implement different GNN varieties [38, 39]. We first consider a multi-layer Graph Convolutional Network (GCN) [40], with three hidden graph convolutional layers of dimension 64. After the first two hidden layers we apply graph normalization [41] over individual graphs and then ReLU activation functions. After the final hidden layer we apply global max pooling, a permutation-invariant aggregator. This outputs a single scalar, our regression target prediction. We then employ Graph Attention Networks (GATs) [42], which leverage masked self-attentional layers. The core architecture is the same; however, we substitute the graph convolutional layer with attentional layers. We also implement a Message Passing Neural Network (MPNN) [31]. This type of architecture has been found specially suitable for molecular property prediction [43]. The model has three hidden message passing layers with `max` aggregation and without graph normalization. The formulation includes permutation-invariant aggregation via global max pooling and a linear prediction head at the end of the network to transform the output message feature vector into a scalar. The MLPs, $\psi$ and $\phi$, are composed of two linear layers with an embedding dimension of 64, 1-dimensional batch normalization, and ReLU activations. We train the networks for 15,000 epochs, with an outer (meta) learning rate of $10^{-3}$, an inner learning rate of $5 \times 10^{-3}$ (for message passing models for QM9 this is reduced to $5 \times 10^{-4}$ to avoid instabilities), $k = 5$ steps of SGD number of internal updates per task, and $K = 10$ samples per task.

### 4.2 Results

Table 1 shows the performance (MSE) with the GCN, GAT and MPNN models for the Alchemy dataset, and Table 2 for the QM9 dataset. The meta-trained models are compared against using a random initialization for the GNN model parameters. As previously mentioned, we train on all but one quantum property and k-shot learn the remaining regression task: in the case of Alchemy we train on 11 and for QM9 on 18. To obtain the mean and standard deviation we calculate the average across all possible tasks, that is, we train 12 models in the case of Alchemy and 19 for QM9. For each meta-trained model we k-shot learn 5 gradient steps (with learning rate equal to the inner learning rate used for training), we do this 100 times, and calculate the overall mean and standard deviation across all tasks. An additional breakdown of all results per task can be found in Appendix B.

**Table 1:** Performance on Alchemy dataset [32]. Comparing $k = 5$-shot optimization across GNN models. K = 10 datapoints (graphs) were used and Reptile was run over 15,000 epochs. Values given are MSE $\pm$ standard deviation (averaged over all tasks excluding *Heat capacity at 298.15 K*, see Appendix B).

| Model | Initialization | Pre-Update | 1 Gradient Step | 5 Gradient Steps |
|---|---|---|---|---|
| GCN | Random | 2.42e+0 ($\pm$ 3.83e-1) | 7.93e-1 ($\pm$1.41e-1) | 1.94e-1 ($\pm$4.46e-2) |
| GAT | Random | 1.21e+0 ($\pm$ 3.34e-1) | 5.57e-1 ($\pm$1.64e-1) | 1.12e-1 ($\pm$3.97e-2) |
| MPNN | Random | 2.44e+0 ($\pm$ 4.86e-1) | 3.19e-1 ($\pm$1.77e-1) | 9.04e-2 ($\pm$8.39e-2) |
| GCN | Meta-Learning | 3.70e-1 ($\pm$ 9.65e-2) | 2.15e-2 ($\pm$ 1.77e-2) | 1.51e-2 ($\pm$ 8.32e-3) |
| GAT | Meta-Learning | 3.21e-1 ($\pm$ 6.73e-2) | 3.88e-2 ($\pm$ 4.12e-2) | 1.43e-2 ($\pm$ 1.36e-2) |
| MPNN | Meta-Learning | 2.80e-1 ($\pm$ 5.50e-2) | 1.74e-2 ($\pm$ 1.42e-2) | 1.35e-2 ($\pm$ 1.30e-2) |

**Table 2:** Performance on QM9 dataset [44, 45]. Comparing $k = 5$-shot optimization across GNN models. K = 10 datapoints (graphs) were used and Reptile was run over 15,000 epochs. Values given are MSE $\pm$ standard deviation (averaged over all tasks).

| Model | Initialization | Pre-Update | 1 Gradient Step | 5 Gradient Steps |
|---|---|---|---|---|
| GCN | Random | 5.21e+0 ($\pm$ 5.32e-1) | 2.89e+0 ($\pm$4.44e-1) | 7.06e-1 ($\pm$8.48e-2) |
| GAT | Random | 2.99e+0 ($\pm$ 3.98e-1) | 2.06e+0 ($\pm$3.13e-1) | 4.23e-1 ($\pm$8.13e-2) |
| MPNN | Random | 2.37e+0 ($\pm$ 4.02e-1) | 5.77e-1 ($\pm$3.25e-1) | 3.28e-1 ($\pm$2.33e-1) |
| GCN | Meta-Learning | 1.14e0 ($\pm$ 9.52e-2) | 2.40e-2 ($\pm$ 2.28e-2) | 1.33e-2 ($\pm$ 8.47e-3) |
| GAT | Meta-Learning | 1.20e0 ($\pm$ 1.34e-1) | 3.15e-2 ($\pm$ 3.20e-2) | 1.20e-2 ($\pm$ 1.03e-2) |
| MPNN | Meta-Learning | 1.29e0 ($\pm$ 8.06e-2) | 9.16e-3 ($\pm$ 6.08e-3) | 6.16e-3 ($\pm$ 4.72e-3) |

These results show that meta-learning algorithms are applicable to graph representation learning and that they can achieve quality results on the prediction of chemical properties. Furthermore, models that make use of more flexible layer types showcase improved performance. Crucially, this finding is replicated across both the Alchemy and QM9 datasets. MPNNs are able to compute messages in the form of vectors based on the feature information of neighboring nodes. We find that this allows the network to more quickly adapt to new tasks during few-shot learning, as compared to GCNs and GATs which use a single scalar to model interactions between nodes.

### 4.3 Ensemble Methods

We further experiment with *ensemble*-based methods which combine the predictions of the meta-learned models for more robust, bolstered generalization for the QM9 dataset [46]. In particular, we use ensembles of meta-learned MPNNs [47], where the number of models we aggregate ranges from $2$ to $4$. Further, we consider two forms of such aggregation, namely, taking a simple average versus learning a weighted sum. Learning a weighted sum will afford improved performance, as the model can learn to adjust and balance contributions from different pre-trained models during few-shot learning. Note that we start few-short learning with the weighting factors initialized uniformly (e.g., to $\frac{1}{M}$, where $M$ is the number of models in our ensemble). Indeed, in Table 3, we find that the weighted sum approach yields better performance. Since the combination is explicitly optimized over, we reason that such results occur, in part, due to the ability of the weighted sum to capture interactions between the models. Also, we highlight that, even before few-shot learning, taking a simple average over the predictions, provided we have several models, confers performance gains on top of a single model, as shown in the *Pre-Update* column in Table 3.

## 5 Conclusion

In this work we have shown the applicability of the Reptile model-agnostic algorithm for meta-learning to GNN based regression tasks. More specifically, we have demonstrated that it is possible to meta-learn across different molecular chemical properties by exploiting the underlying graph structure. We have experimentally shown that providing models with more expressive GNN layers leads to improved performance and that ensemble-methods can also be beneficial for meta-learning. Note that in Appendix D we have included some additional ensemble experiments using equivariant GNN layers given the recent success of architectures that exploit equivariance and invariance in the literature [47–49].

**Table 3:** MPNN ensemble performance on QM9 dataset [44, 45] using Reptile [3]. Values given are MSE $\pm$ standard deviation. These results are only testing on the *Dipole moment* and using MPNN models.

| No. Models ($M$) | Initialization | Agg Method | Pre-Update | 1 Gradient Step | 5 Gradient Steps |
|---|---|---|---|---|---|
| 1 | Random | N/A | 5.47e-1 ($\pm$ 2.33e-1) | 3.52e-1 ($\pm$ 3.29e-1) | 3.19e-1 ($\pm$ 2.16e-1) |
| 1 | Meta-learning | N/A | 3.82e-1 ($\pm$ 2.10e-2) | 1.33e-3 ($\pm$ 1.16e-3) | 2.98e-4 ($\pm$ 2.18e-4) |
| 2 | Meta-learning | Average | 8.07e-4 ($\pm$ 3.13e-3) | 3.35e-4 ($\pm$ 7.25e-4 ) | 1.77e-4 ($\pm$ 8.95e-5) |
| 3 | Meta-learning | Average | 3.38e-4 ($\pm$ 5.43e-4) | 2.34e-4 ($\pm$ 2.49e-4) | 1.45e-4 ($\pm$ 7.71e-5) |
| 4 | Meta-learning | Average | 2.58e-4 ($\pm$ 9.70e-4) | 3.01e-4 ($\pm$ 2.80e-2) | 1.24e-4 ($\pm$ 7.43e-5) |
| 2 | Meta-learning | Learned | 8.07e-4 ($\pm$ 3.13e-3) | 2.48e-4 ($\pm$ 1.35e-4) | 1.24e-4 ($\pm$ 6.14e-5) |
| 3 | Meta-learning | Learned | 3.38e-4 ($\pm$ 5.43e-4) | 2.23e-4 ($\pm$ 3.41e-4) | 1.20e-4 ($\pm$ 2.83e-4) |
| 4 | Meta-learning | Learned | 2.58e-4 ($\pm$ 9.70e-4) | 1.80e-4 ($\pm$ 5.44e-4) | 8.04e-5 ($\pm$ 4.42e-5) |

As part of future research, it would be interesting to take into account field knowledge: in this experiments we have meta-learned across all available molecular properties, it might be better to meta-learn only on some particular molecular properties depending on the task for which we want to k-shot learn during testing.

## Acknowledgements

We would like to thank the Rafael del Pino Foundation for supporting Haitz Sáez de Ocáriz Borde's studies at the University of Cambridge.

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

## A   Further Discussion on Message Passing Expressivity

In this section we give further insights into message passing expressivity. In this work, we refer to expressivity as the ability of GNN layers to flexibly share information between adjacent nodes in the graph. The MPNN model mentioned in the main text shares information between nodes by calculating non-linear mappings of the node neighbor features according to the full expression

$$\mathbf{h}_i^{(l)} = \phi \left( \mathbf{h}_i^{(l-1)}, \bigoplus_{j \in \mathcal{N}_i} \psi(\mathbf{h}_i^{(l-1)}, \mathbf{h}_j^{(l-1)}, e_{ij}) \right),$$

which was previously introduced in Section 2.2. $\psi$ is a MLP which in principle is a universal approximator and could approximate any arbitrary function given the network has enough capacity. Hence, we say this construction is the most expressive, or flexible. On the other hand, attentional models learn different learnable coefficients for each neighbor and use these to update the node features [42]. This is less flexible than using a fully non-linear mapping as before. Lastly, convolutional models use the same weighting for all nodes in the same neighborhood [40], and hence they are even less expressive because they cannot consider the contribution of different nodes in isolation, or pay more attention to specific nodes.

## B   Results Breakdown per Task

In Table 4, Table 5 and Table 6 we provide the k-shot learning results for each regression task and GNN model. From the tables, it is clear that meta-learning accelerates learning new molecular regression tasks as compared to the randomly initialized GCN, GAT, and MPNN baselines.

In Table 4 we can see that the only property regression task that does not benefit substantially from meta-learning is the *Heat capacity at 298.15 K*. The reason behind it remains unclear. We hypothesize that *Heat capacity at 298.15 K* may not be as closely related to the rest of the molecular properties for the algorithm to meta-learn successfully. As discussed in Section 5, considering field knowledge could improve the performance. This might be done by only meta-learning based on tasks that are most closely related or that share physical mechanisms with the *Heat capacity at 298.15 K* of the molecules.

Also, in the case of Alchemy note that although increased expressivity in the GNN models is clearly helpful for testing on properties such as the *Dipole moment*, *Polarizability*, *Highest occupied molecular orbital energy*, *Gap*, *Enthalpy at 298.15 K*, and *Free energy at 298.15 K*, it is not so obviously the case for other properties like *Lowest unoccupied molecular orbital energy*, *R2*, *Internal energy*, and *Internal energy at 298.15 K*, and in these, performance may be highly dependent on network initialization. In Table 6, for the QM9 dataset there is a more clear correlation between increased network expressivity and improved meta-learning performance when applying k-shot learning for new regression tasks; nevertheless, it is still possible to find a few exceptions.

Lastly, as previously mentioned in the main text, the internal learning rate and k-shot learning rate for convolutional and attentional models is of $5 \times 10^{-3}$, whereas for message passing models we use $5 \times 10^{-4}$. This is because the message passing models struggle to converge for larger learning rates.

## C   Further Details on Training and Testing Procedures

In this section we provide further clarifications regarding the training procedure, normalization of the data, and splits. We split the datasets into train and test set. For training we use $90\%$ of the molecules available in the dataset, and the remaining $10\%$ are used for testing. The splits are random. During training the models are trained to meta-learn across all but one task. For testing, we use new unseen molecules from the test set and k-shot learn also on a new molecular property regression task, which the model has never seen before.

This may more clearly be illustrated using an example. Let us refer back to Table 4, and focus on the first row in which we apply meta-learning (row 38 counting the header as a row). The task is to k-shot learn the *Dipole moment*. To do so, we use a GCN whose weights have been pretrained using meta-learning. This model has been trained by being fed molecules from the train split and applying meta-learning across all task but the *Dipole moment*. That is, it has been trained to predict the *Polarizability*, the *Highest occupied molecular orbital energy*, the *Lowest unoccupied molecular orbital energy*, the *Gap*, the *R2*, the *Zero point energy*, the *Internal energy*, the *Internal energy at 298.15 K*, the *Enthalpy at 298.15 K*, the *Free energy at 298.15 K*, and the *Heat capacity at 298.15 K*. Once pretrained using meta-learning we k-shot learn based on a new set of molecules (the ones from the test set). Apart from working with previously unseen molecules we also try to predict a new task:

**Table 4:** Performance on Alchemy dataset [32]. In this table we provide a breakdown of the performance across all tasks. K = 10 datapoints (graphs) were used and Reptile was run over 15,000 epochs. Values given are MSE ± standard deviation.

| Model | Initialization | Task | Pre-Update | 1 Gradient Step | 5 Gradient Steps |
|---|---|---|---|---|---|
| GCN | Random | Dipole moment | 2.41e+0 (± 6.12e-1) | 3.08e-1 (±1.27e-1) | 2.72e-2 (±1.11e-2) |
| GCN | Random | Polarizability | 5.10e+0 (± 9.81e-1) | 1.63e+0 (±3.68e-1) | 1.91e-1 (±3.22e-2) |
| GCN | Random | Highest occupied molecular orbital energy | 1.25e+0 (± 4.04e-1) | 1.61e-1 (±7.07e-2) | 2.55e-2 (±1.03e-2) |
| GCN | Random | Lowest unoccupied molecular orbital energy | 5.49e-1 (± 2.12e-1) | 1.90e-1 (±9.46e-2) | 7.52e-2 (±4.65e-2) |
| GCN | Random | Gap | 4.16e-1 (± 3.03e-1) | 7.79e-2 (±3.08e-2) | 2.01e-2 (±6.83e-3) |
| GCN | Random | R2 | 7.69e-1 (± 2.63e-1) | 4.63e-1 (±1.74e-1) | 1.69e-1 (±6.93e-2) |
| GCN | Random | Zero point energy | 2.96e-1 (± 1.10e-1) | 1.18e-1 (±4.97e-2) | 3.55e-2 (±1.86e-2) |
| GCN | Random | Internal energy | 1.04e+0 (± 2.08e-1) | 5.76e-1 (±1.29e-1) | 2.14e-1 (±6.73e-2) |
| GCN | Random | Internal energy at 298.15 K | 4.70e+0 (± 6.74e-1) | 2.49e+0 (±4.94e-1) | 3.65e-1 (±9.84e-2) |
| GCN | Random | Enthalpy at 298.15 K | 7.50e-2 (± 4.20e-2) | 3.80e-2 (±1.95e-2) | 1.53e-2 (±8.40e-3) |
| GCN | Random | Free energy at 298.15 K | 3.09e-1 (± 8.66e-2) | 6.72e-2 (±3.17e-2) | 1.77e-2 (±1.05e-2) |
| GCN | Random | Heat capacity at 298.15 K | 6.97e+0 (± 1.13e+0) | 4.24e+0 (±1.14e+0) | 1.30e+0 (±5.79e-1) |
| GAT | Random | Dipole moment | 1.35e-1 (± 5.47e-2) | 9.19e-2 (±3.96e-2) | 3.06e-2 (±1.74e-2) |
| GAT | Random | Polarizability | 7.49e-1 (± 2.12e-1) | 1.40e-1 (±4.45e-2) | 3.41e-2 (±1.55e-2) |
| GAT | Random | Highest occupied molecular orbital energy | 1.95e+0 (± 6.49e-1) | 3.01e-1 (±1.19e-1) | 3.23e-2 (±1.14e-2) |
| GAT | Random | Lowest unoccupied molecular orbital energy | 4.17e+0 (± 1.30e+0) | 2.22e+0 (±5.67e-1) | 5.68e-1 (±9.30e-2) |
| GAT | Random | Gap | 1.88e-1 (± 7.61e-2) | 1.12e-1 (±1.15e-1) | 2.80e-2 (±1.73e-2) |
| GAT | Random | R2 | 4.19e-1 (± 2.00e-1) | 2.11e-1 (±9.02e-2) | 8.75e-2 (±4.63e-2) |
| GAT | Random | Zero point energy | 5.67e+0 (± 1.17e+0) | 2.22e-1 (±2.23e-1) | 2.79e-2 (±1.65e-2) |
| GAT | Random | Internal energy | 1.11e+0 (± 2.21e-1) | 5.91e-1 (±1.92e-1) | 2.24e-1 (±9.30e-2) |
| GAT | Random | Internal energy at 298.15 K | 9.66e-1 (± 4.06e-1) | 6.35e-1 (±1.68e-1) | 1.71e-1 (±5.91e-2) |
| GAT | Random | Enthalpy at 298.15 K | 7.88e-1 (± 2.91e-1) | 1.56e-1 (±9.67e-2) | 1.96e-2 (±1.34e-2) |
| GAT | Random | Free energy at 298.15 K | 3.35e+0 (± 7.13e-1) | 1.31e+0 (±2.36e-1) | 2.08e-1 (±4.03e-2) |
| GAT | Random | Heat capacity at 298.15 K | 4.78e+0 (± 1.12e+0) | 2.11e+0 (±1.16e+0) | 8.00e-1 (±5.51e-1) |
| MPNN | Random | Dipole moment | 4.71e-1 (± 2.09e-1) | 2.03e-1 (±1.18e-1) | 7.01e-2 (±7.86e-2) |
| MPNN | Random | Polarizability | 1.41e+1 (± 1.35e+0) | 9.28e-1 (±3.88e-1) | 6.16e-2 (±6.13e-2) |
| MPNN | Random | Highest occupied molecular orbital energy | 3.64e-1 (± 1.95e-1) | 1.46e-1 (±9.14e-2) | 5.58e-2 (±6.01e-2) |
| MPNN | Random | Lowest unoccupied molecular orbital energy | 1.60e+0 (± 4.05e-1) | 4.84e-1 (±2.12e-1) | 1.59e-1 (±1.10e-1) |
| MPNN | Random | Gap | 5.70e-1 (± 4.23e-1) | 3.48e-1 (±2.80e-1) | 1.54e-1 (±1.88e-1) |
| MPNN | Random | R2 | 4.25e+0 (± 7.03e-1) | 2.65e-1 (±1.24e-1) | 5.61e-2 (±5.12e-2) |
| MPNN | Random | Zero point energy | 7.97e+0 (± 9.52e-1) | 8.96e-1 (±2.81e-1) | 7.36e-2 (±9.39e-2) |
| MPNN | Random | Internal energy | 6.22e-1 (± 2.84e-1) | 2.76e-1 (±1.42e-1) | 1.47e-1 (±8.96e-2) |
| MPNN | Random | Internal energy at 298.15 K | 5.07e+0 (± 8.66e-1) | 5.37e-1 (±2.41e-1) | 9.73e-2 (±7.07e-2) |
| MPNN | Random | Enthalpy at 298.15 K | 2.86e+0 (± 5.90e-1) | 2.95e-1 (±1.78e-1) | 5.46e-2 (±7.02e-2) |
| MPNN | Random | Free energy at 298.15 K | 1.97e+0 (± 4.95e-1) | 3.57e-1 (±2.29e-1) | 9.47e-2 (±1.24e-1) |
| MPNN | Random | Heat capacity at 298.15 K | 1.79e+1 (± 2.06e+0) | 3.49e+0 (±9.03e-1) | 5.77e-1 (±3.71e-1) |
| GCN | Meta-learning | Dipole moment | 1.41e-2 (± 1.40e-2) | 4.27e-3 (± 5.32e-3) | 1.82e-3 (± 1.96e-3) |
| GCN | Meta-learning | Polarizability | 6.49e-3 (± 6.52e-3) | 1.70e-3 (± 2.21e-3) | 7.52e-4 (± 1.22e-3) |
| GCN | Meta-learning | Highest occupied molecular orbital energy | 2.41e-3 (± 2.59e-3) | 1.52e-3 (± 1.89e-3) | 9.97e-4 (± 1.32e-3) |
| GCN | Meta-learning | Lowest unoccupied molecular orbital energy | 7.41e-1 (± 1.31e-1) | 4.42e-2 (± 3.80e-2) | 4.32e-2 (± 2.19e-2) |
| GCN | Meta-learning | Gap | 1.99e-2 (± 1.12e-2) | 4.25e-3 (± 3.74e-3) | 2.29e-3 (± 1.38e-3) |
| GCN | Meta-learning | R2 | 6.79e-1 (± 1.37e-1) | 5.07e-2 (± 3.17e-2) | 4.16e-2 (± 1.71e-2) |
| GCN | Meta-learning | Zero point energy | 2.13e-2 (± 6.32e-3) | 2.72e-3 (± 2.85e-3) | 1.58e-3 (± 1.68e-3) |
| GCN | Meta-learning | Internal energy | 1.27e+0 (± 1.92e-1) | 4.79e-2 (± 3.98e-2) | 4.05e-2 (± 1.79e-2) |
| GCN | Meta-learning | Internal energy at 298.15 K | 1.18e+0 (± 2.26e-1) | 7.09e-2 (± 6.26e-2) | 3.00e-2 (± 2.46e-2) |
| GCN | Meta-learning | Enthalpy at 298.15 K | 9.96e-2 (± 2.01e-2) | 5.25e-3 (± 3.20e-3) | 1.58e-3 (± 9.02e-4) |
| GCN | Meta-learning | Free energy at 298.15 K | 3.94e-2 (± 1.82e-2) | 3.60e-3 (± 3.40e-3) | 1.51e-3 (± 1.55e-3) |
| GCN | Meta-learning | Heat capacity at 298.15 K | 1.07e+1 (± 1.48e+0) | 6.44e+0 (± 1.05e+0) | 1.60e+0 (± 0.43e+0) |
| GAT | Meta-learning | Dipole moment | 1.11e-1 (± 3.71e-2) | 5.16e-3 (± 3.66e-3) | 1.18e-4 (± 2.32e-4) |
| GAT | Meta-learning | Polarizability | 2.98e-3 (± 4.15e-3) | 4.99e-4 (± 8.48e-4) | 6.55e-5 (± 2.92e-4) |
| GAT | Meta-learning | Highest occupied molecular orbital energy | 3.53e-2 (± 1.68e-2) | 1.28e-3 (± 3.41e-3) | 2.78e-4 (± 1.87e-3) |
| GAT | Meta-learning | Lowest unoccupied molecular orbital energy | 1.08e+0 (± 1.74e-1) | 1.08e-1 (± 9.91e-2) | 4.25e-2 (± 4.13e-2) |
| GAT | Meta-learning | Gap | 5.63e-3 (± 4.36e-3) | 1.00e-3 (± 2.69e-3) | 2.77e-4 (± 6.76e-4) |
| GAT | Meta-learning | R2 | 7.42e-1 (± 1.62e-1) | 5.92e-2 (± 4.60e-2) | 4.20e-2 (± 3.81e-2) |
| GAT | Meta-learning | Zero point energy | 4.48e-2 (± 2.32e-2) | 2.51e-3 (± 1.04e-2) | 7.86e-4 (± 3.94e-3) |
| GAT | Meta-learning | Internal energy | 9.10e-1 (± 1.72e-1) | 1.71e-1 (± 2.41e-1) | 4.23e-2 (± 3.85e-2) |
| GAT | Meta-learning | Internal energy at 298.15 K | 5.83e-1 (± 1.21e-1) | 7.59e-2 (± 3.62e-2) | 2.82e-2 (± 2.25e-2) |
| GAT | Meta-learning | Enthalpy at 298.15 K | 1.08e-2 (± 2.02e-2) | 2.29e-3 (± 9.78e-3) | 3.04e-4 (± 2.06e-3) |
| GAT | Meta-learning | Free energy at 298.15 K | 9.84e-3 (± 5.10e-3) | 4.25e-4 (± 4.80e-4 ) | 2.11e-5 (± 1.26e-5) |
| GAT | Meta-learning | Heat capacity at 298.15 K | 9.92e+0 (± 1.47e+0) | 7.49e+0 (± 1.31e+0) | 2.33e+0 (± 9.69e-1) |
| MPNN | Meta-learning | Dipole moment | 9.86e-2 (± 6.96e-3) | 1.88e-3 (± 6.53e-4) | 5.81e-5 (± 5.05e-5) |
| MPNN | Meta-learning | Polarizability | 5.62e-2 (± 6.09e-3) | 2.58e-4 (± 2.88e-4) | 1.72e-5 (± 1.26e-5) |
| MPNN | Meta-learning | Highest occupied molecular orbital energy | 1.38e-3 (± 1.15e-3 ) | 6.50e-5 (± 3.56e-5) | 5.55e-5 (± 2.69e-5) |
| MPNN | Meta-learning | Lowest unoccupied molecular orbital energy | 9.15e-1 (± 1.42e-1) | 4.43e-2 (± 3.29e-2) | 3.76e-2 (± 3.21e-2) |
| MPNN | Meta-learning | Gap | 2.07e-3 (± 6.30e-4 ) | 3.36e-5 (± 5.62e-5) | 2.86e-5 (± 5.07e-5) |
| MPNN | Meta-learning | R2 | 5.46e-1 (± 1.44e-1 ) | 4.39e-2 (± 4.37e-2) | 4.20e-2 (± 4.34e-2) |
| MPNN | Meta-learning | Zero point energy | 1.27e-1 (± 7.92e-3) | 1.64e-3 (± 1.37e-3) | 3.99e-4 (± 2.36e-4) |
| MPNN | Meta-learning | Internal energy | 4.15e-1 (± 1.26e-1) | 5.27e-2 (± 3.98e-2) | 4.01e-2 (± 3.99e-2) |
| MPNN | Meta-learning | Internal energy at 298.15 K | 8.32e-1 (± 1.58e-1) | 4.48e-2 (± 3.69e-2) | 2.82e-2 (± 2.67e-2) |
| MPNN | Meta-learning | Enthalpy at 298.15 K | 1.89e-2 (± 1.83e-3) | 7.05e-5 (± 6.79e-5) | 1.49e-5 (± 9.44e-6) |
| MPNN | Meta-learning | Free energy at 298.15 K | 7.09e-2 (± 1.09e-2) | 1.34e-3 (± 8.02e-4 ) | 2.29e-5 (± 1.95e-5) |
| MPNN | Meta-learning | Heat capacity at 298.15 K | 1.02e+1 (± 1.21e+0) | 4.06e-1 (± 1.97e-1) | 3.47e-1 (± 1.85e-1) |

**Table 5:** Performance on QM9 dataset [32] using randomly initialized networks. In this table we provide a breakdown of the performance across all tasks. K = 10 datapoints (graphs) were used and Reptile was run over 15,000 epochs. Values given are MSE ± standard deviation.

| Model | Initialization | Task | Pre-Update | 1 Gradient Step | 5 Gradient Steps |
|---|---|---|---|---|---|
| GCN | Random | Dipole moment | 1.75e-1 (± 5.55e-2) | 9.52e-2 (±4.48e-2) | 3.76e-2 (±2.70e-2) |
| GCN | Random | Isotropic polarizability | 5.54e-1 (± 1.46e-1) | 3.13e-1 (±1.31e-1) | 8.65e-2 (±6.21e-2) |
| GCN | Random | Highest occupied molecular orbital energy | 1.13e+0 (± 3.29e-1) | 8.92e-2 (±5.79e-2) | 1.63e-2 (±8.31e-3) |
| GCN | Random | Lowest unoccupied molecular orbital energy | 8.44e-1 (± 2.76e-1) | 2.68e-1 (±1.18e-1) | 1.32e-2 (±5.47e-3) |
| GCN | Random | Gap | 3.48e-1 (± 1.05e-1) | 3.02e-1 (±1.10e-1) | 1.53e-1 (±5.72e-2) |
| GCN | Random | R2 | 1.72e-1 (± 6.57e-2) | 6.10e-2 (±2.61e-2) | 1.65e-2 (±7.67e-3) |
| GCN | Random | Zero point vibrational energy | 6.62e-1 (± 1.10e-1) | 3.70e-1 (±8.35e-2) | 4.17e-2 (±8.25e-3) |
| GCN | Random | Internal energy at 0K | 1.54e+1 (± 1.25e+0) | 1.13e+1 (±1.64e+0) | 2.15e+0 (±2.40e-1) |
| GCN | Random | Internal energy at 298.15K | 9.47e+0 (± 8.68e-1) | 6.76e+0 (±8.00e-1) | 1.98e+0 (±1.90e-1) |
| GCN | Random | Enthalpy at 298.15K | 1.98e+1 (± 2.08e+0) | 8.51e+0 (±1.44e+0) | 1.84e+0 (±1.87e-1) |
| GCN | Random | Free energy at 298.15K | 1.36e+1 (± 1.03e+0) | 6.35e+0 (±8.25e-1) | 2.05e+0 (±1.90e-1) |
| GCN | Random | Heat capacity at 298.15K | 4.08e-1 (± 7.03e-2) | 2.86e-1 (±5.93e-2) | 8.65e-2 (±3.38e-2) |
| GCN | Random | Atomization energy at 0K | 2.17e+0 (± 3.57e-1) | 7.96e-1 (±1.26e-1) | 9.08e-2 (±1.58e-2) |
| GCN | Random | Atomization energy at 298.15K | 2.23e-1 (± 1.24e-1) | 2.81e-2 (±1.66e-2) | 1.33e-2 (±7.72e-3) |
| GCN | Random | Atomization enthalpy at 298.15K | 3.63e-1 (± 1.56e-1) | 2.68e-1 (±1.10e-1) | 1.08e-1 (±5.00e-2) |
| GCN | Random | Atomization free energy at 298.15K | 1.54e-1 (± 7.19e-2) | 8.73e-2 (±6.02e-2) | 3.13e-2 (±2.10e-2) |
| GCN | Random | Rotational constant A | 1.39e+0 (± 4.15e-1) | 1.23e-1 (±6.98e-2) | 1.30e-2 (±6.89e-3) |
| GCN | Random | Rotational constant B | 7.38e-1 (± 9.70e-2) | 4.52e-1 (±1.30e-1) | 1.85e-1 (±7.38e-2) |
| GCN | Random | Rotational constant C | 4.18e-2 (± 1.66e-2) | 3.24e-2 (±1.17e-2) | 1.72e-2 (±5.55e-3) |
| GAT | Random | Dipole moment | 1.03e-1 (± 5.79e-2) | 3.81e-2 (±2.24e-2) | 9.47e-3 (±7.33e-3) |
| GAT | Random | Isotropic polarizability | 4.49e-1 (± 1.73e-1) | 6.21e-2 (±4.06e-2) | 7.24e-3 (±5.70e-3) |
| GAT | Random | Highest occupied molecular orbital energy | 3.56e-1 (± 4.28e-1) | 1.12e-1 (±6.53e-2) | 1.03e-2 (±7.90e-3) |
| GAT | Random | Lowest unoccupied molecular orbital energy | 7.71e-1 (± 2.05e-1) | 8.96e-2 (±5.81e-2) | 1.37e-2 (±7.86e-3) |
| GAT | Random | Gap | 2.55e-1 (± 1.11e-1) | 1.16e-1 (±5.14e-2) | 3.44e-2 (±2.48e-2) |
| GAT | Random | R2 | 4.02e-1 (± 3.17e-1) | 1.10e-1 (±6.80e-2) | 2.94e-2 (±1.78e-2) |
| GAT | Random | Zero point vibrational energy | 1.07e-1 (± 4.01e-2) | 6.44e-2 (±2.37e-2) | 2.24e-2 (±1.21e-2) |
| GAT | Random | Internal energy at 0K | 1.43e+1 (± 1.42e+0) | 1.24e+1 (±1.60e+0) | 1.87e+0 (±3.45e-1) |
| GAT | Random | Internal energy at 298.15K | 4.70e+0 (± 4.61e-1) | 2.80e+0 (±3.40e-1) | 9.11e-1 (±1.54e-1) |
| GAT | Random | Enthalpy at 298.15K | 3.19e+0 (± 3.68e-1) | 2.00e+0 (±3.13e-1) | 4.41e-1 (±1.23e-1) |
| GAT | Random | Free energy at 298.15K | 1.07e+1 (± 7.29e-1) | 6.59e+0 (±1.02e+0) | 1.61e+0 (±2.13e-1) |
| GAT | Random | Heat capacity at 298.15K | 4.69e-1 (± 4.59e-1) | 2.93e-1 (±1.51e-1) | 1.17e-1 (±5.83e-2) |
| GAT | Random | Atomization energy at 0K | 6.87e+0 (± 1.02e+0) | 7.45e-1 (±3.72e-1) | 2.61e-2 (±1.62e-2) |
| GAT | Random | Atomization energy at 298.15K | 1.71e-1 (± 8.63e-2) | 1.01e-1 (±5.66e-2) | 3.06e-2 (±1.39e-2) |
| GAT | Random | Atomization enthalpy at 298.15K | 3.37e+0 (± 7.45e-1) | 4.62e-1 (±2.93e-1) | 1.82e-2 (±9.31e-3) |
| GAT | Random | Atomization free energy at 298.15K | 1.42e+0 (± 4.73e-1) | 1.46e-1 (±2.16e-1) | 4.13e-2 (±2.53e-2) |
| GAT | Random | Rotational constant A | 1.34e+0 (± 5.38e-1) | 1.46e-1 (±6.62e-2) | 3.27e-2 (±2.30e-2) |
| GAT | Random | Rotational constant B | 2.86e-1 (± 1.23e-1) | 9.56e-2 (±6.73e-2) | 2.32e-2 (±2.18e-2) |
| GAT | Random | Rotational constant C | 3.52e+0 (± 7.03e-1) | 1.30e-1 (±1.32e-1) | 1.63e-2 (±9.44e-3) |
| MPNN | Random | Dipole moment | 5.47e-1 (± 2.33e-1) | 3.52e-1 (±3.29e-1) | 3.19e-1 (±2.16e-1) |
| MPNN | Random | Isotropic polarizability | 3.85e-1 (± 1.33e-1) | 1.42e-1 (±8.76e-2) | 1.73e-1 (±1.27e-1) |
| MPNN | Random | Highest occupied molecular orbital energy | 1.10e+0 (± 2.91e-1) | 6.62e-1 (±3.10e-1) | 4.61e-1 (±2.97e-1) |
| MPNN | Random | Lowest unoccupied molecular orbital energy | 4.16e-1 (± 1.67e-1) | 3.21e-1 (±1.60e-1) | 3.71e-1 (±2.27e-1) |
| MPNN | Random | Gap | 2.62e+0 (± 7.05e-1) | 1.12e+0 (±8.07e-1) | 8.21e-1 (±5.53e-1) |
| MPNN | Random | R2 | 8.55e-1 (± 2.02e-1) | 5.07e-1 (±2.61e-1) | 2.73e-1 (±2.08e-1) |
| MPNN | Random | Zero point vibrational energy | 1.66e+0 (± 3.44e-1) | 6.20e-1 (±2.55e-1) | 1.27e-1 (±1.07e-1) |
| MPNN | Random | Internal energy at 0K | 1.17e+0 (± 2.83e-1) | 4.63e-1 (±2.15e-1) | 2.28e-1 (±1.55e-1) |
| MPNN | Random | Internal energy at 298.15K | 1.37e+0 (± 3.40e-1) | 4.97e-1 (±2.58e-1) | 2.73e-1 (±2.04e-1) |
| MPNN | Random | Enthalpy at 298.15K | 3.05e+0 (± 5.55e-1) | 4.91e-1 (±2.28e-1) | 1.53e-1 (±1.55e-1) |
| MPNN | Random | Free energy at 298.15K | 3.44e+0 (± 6.15e-1) | 1.05e+0 (±5.74e-1) | 5.47e-1 (±3.84e-1) |
| MPNN | Random | Heat capacity at 298.15K | 1.19e+1 (± 9.56e-1) | 6.99e-1 (±4.17e-1) | 1.89e-1 (±1.65e-1) |
| MPNN | Random | Atomization energy at 0K | 6.44e+0 (± 6.49e-1) | 3.40e-1 (±1.98e-1) | 1.74e-1 (±1.60e-1) |
| MPNN | Random | Atomization energy at 298.15K | 3.50e-1 (± 1.59e-1) | 2.96e-1 (±2.26e-1) | 5.15e-1 (±4.13e-1) |
| MPNN | Random | Atomization enthalpy at 298.15K | 2.16e-1 (± 1.00e-1) | 1.80e-1 (±1.31e-1) | 7.14e-1 (±5.32e-1) |
| MPNN | Random | Atomization free energy at 298.15K | 3.91e+0 (± 4.81e-1) | 6.00e-1 (±2.86e-1) | 2.52e-1 (±2.41e-1) |
| MPNN | Random | Rotational constant A | 2.78e+0 (± 4.76e-1) | 1.61e+0 (±7.41e-1) | 3.41e-1 (±2.77e-1) |
| MPNN | Random | Rotational constant B | 7.07e-1 (± 3.20e-1) | 4.28e-1 (±2.31e-1) | 1.74e-1 (±1.58e-1) |
| MPNN | Random | Rotational constant C | 9.61e+0 (± 1.07e+0) | 1.48e+0 (±1.08e+0) | 9.79e-1 (±7.31e-1) |

**Table 6:** Performance on QM9 dataset [32] using meta-learning. In this table we provide a breakdown of the performance across all tasks. K = 10 datapoints (graphs) were used and Reptile was run over 15,000 epochs. Values given are MSE ± standard deviation.

| Model | Initialization | Task | Pre-Update | 1 Gradient Step | 5 Gradient Steps |
|---|---|---|---|---|---|
| GCN | Meta-learning | Dipole moment | 1.82e-1 (± 1.51e-2) | 4.30e-3 (± 3.48e-3) | 1.01e-3 (± 1.03e-3) |
| GCN | Meta-learning | Isotropic polarizability | 4.10e-1 (± 4.58e-2) | 3.39e-3 (± 3.77e-3) | 1.37e-3 (± 1.10e-3) |
| GCN | Meta-learning | Highest occupied molecular orbital energy | 2.34e-1 (± 2.30e-2) | 4.69e-3 (± 4.02e-3) | 1.87e-3 (± 9.94e-4) |
| GCN | Meta-learning | Lowest unoccupied molecular orbital energy | 1.92e-1 (± 1.06e-2) | 7.50e-3 (± 5.28e-3) | 5.75e-4 (± 4.48e-4) |
| GCN | Meta-learning | Gap | 1.88e-1 (± 1.08e-2) | 2.58e-3 (± 2.21e-3) | 7.14e-4 (± 1.36e-3) |
| GCN | Meta-learning | R2 | 4.41e-1 (± 4.44e-2) | 2.20e-2 (± 1.16e-2) | 9.12e-3 (± 3.62e-3) |
| GCN | Meta-learning | Zero point vibrational energy | 5.31e-2 (± 1.10e-2) | 2.29e-3 (± 1.65e-3) | 1.27e-3 (± 8.27e-4) |
| GCN | Meta-learning | Internal energy at 0K | 3.87e+0 (± 2.97e-1) | 5.99e-2 (± 3.45e-2) | 5.52e-2 (± 3.29e-2) |
| GCN | Meta-learning | Internal energy at 298.15K | 4.27e+0 (± 3.42e-1) | 6.14e-2 (± 3.75e-2) | 5.15e-2 (± 3.63e-2) |
| GCN | Meta-learning | Enthalpy at 298.15K | 5.27e+0 (± 3.54e-1) | 6.14e-2 (± 4.21e-2) | 5.41e-2 (± 3.77e-2) |
| GCN | Meta-learning | Free energy at 298.15K | 3.98e+0 (± 3.87e-1) | 8.49e-2 (± 1.39e-1) | 5.30e-2 (± 2.77e-2) |
| GCN | Meta-learning | Heat capacity at 298.15K | 3.59e-1 (± 5.13e-2) | 2.48e-2 (± 3.07e-2) | 3.91e-3 (± 2.69e-3) |
| GCN | Meta-learning | Atomization energy at 0K | 2.65e-1 (± 1.64e-2) | 5.68e-3 (± 4.36e-3) | 1.00e-3 (± 7.75e-4) |
| GCN | Meta-learning | Atomization energy at 298.15K | 4.18e-1 (± 3.06e-2) | 1.23e-2 (± 1.28e-2) | 3.68e-3 (± 2.28e-3) |
| GCN | Meta-learning | Atomization enthalpy at 298.15K | 2.04e-1 (± 3.58e-2) | 2.10e-2 (± 5.15e-2) | 5.09e-3 (± 2.09e-3) |
| GCN | Meta-learning | Atomization free energy at 298.15K | 2.35e-1 (± 2.32e-2) | 9.26e-3 (± 6.44e-3) | 2.51e-3 (± 1.27e-3) |
| GCN | Meta-learning | Rotational constant A | 2.56e-1 (± 1.87e-2) | 6.23e-3 (± 9.09e-3) | 8.45e-4 (± 1.08e-3) |
| GCN | Meta-learning | Rotational constant B | 1.96e-1 (± 2.16e-2) | 5.57e-3 (± 6.06e-3) | 9.72e-4 (± 5.73e-4) |
| GCN | Meta-learning | Rotational constant C | 6.71e-1 (± 7.03e-2) | 5.59e-2 (± 2.62e-2) | 5.80e-3 (± 6.10e-3) |
| GAT | Meta-learning | Dipole moment | 1.95e-1 (± 1.06e-2) | 8.00e-3 (± 4.47e-3) | 3.82e-4 (± 6.80e-4) |
| GAT | Meta-learning | Isotropic polarizability | 2.33e-1 (± 2.73e-2) | 5.01e-2 (± 3.52e-2) | 1.29e-3 (± 4.59e-3) |
| GAT | Meta-learning | Highest occupied molecular orbital energy | 9.27e-2 (± 1.49e-1) | 2.44e-2 (± 1.73e-1) | 7.73e-3 (± 5.71e-2) |
| GAT | Meta-learning | Lowest unoccupied molecular orbital energy | 6.76e-1 (± 2.76e-2) | 1.18e-2 (± 1.52e-2) | 1.49e-3 (± 1.25e-3) |
| GAT | Meta-learning | Gap | 3.54e-2 (± 2.69e-2) | 6.32e-3 (± 1.21e-2) | 7.61e-4 (± 2.28e-3) |
| GAT | Meta-learning | R2 | 5.48e-1 (± 8.80e-2) | 1.98e-2 (± 9.98e-3) | 3.95e-3 (± 2.57e-3) |
| GAT | Meta-learning | Zero point vibrational energy | 3.95e-1 (± 5.16e-2) | 3.05e-2 (± 2.13e-2) | 1.08e-4 (± 1.98e-4) |
| GAT | Meta-learning | Internal energy at 0K | 3.18e+0 (± 3.07e-1) | 8.85e-2 (± 4.94e-2) | 5.42e-2 (± 3.01e-2) |
| GAT | Meta-learning | Internal energy at 298.15K | 5.45e+0 (± 3.29e-1) | 7.92e-2 (± 8.86e-2) | 4.74e-2 (± 2.59e-2) |
| GAT | Meta-learning | Enthalpy at 298.15K | 4.63e+0 (± 3.61e-1) | 1.16e-1 (± 5.22e-2) | 4.84e-2 (± 2.37e-2) |
| GAT | Meta-learning | Free energy at 298.15K | 4.72e+0 (± 4.93e-1) | 7.02e-2 (± 3.58e-2) | 5.29e-2 (± 2.65e-2) |
| GAT | Meta-learning | Heat capacity at 298.15K | 2.89e-1 (± 3.68e-2) | 5.45e-3 (± 1.67e-2) | 1.24e-3 (± 1.01e-2) |
| GAT | Meta-learning | Atomization energy at 0K | 2.99e-1 (± 4.72e-1) | 4.62e-2 (± 2.19e-2) | 4.47e-3 (± 1.28e-3) |
| GAT | Meta-learning | Atomization energy at 298.15K | 2.15e-1 (± 1.46e-2) | 2.39e-3 (± 1.26e-2) | 7.12e-4 (± 4.30e-3) |
| GAT | Meta-learning | Atomization enthalpy at 298.15K | 3.41e-1 (± 3.88e-2) | 8.67e-3 (± 9.55e-3) | 8.55e-4 (± 1.84e-3) |
| GAT | Meta-learning | Atomization free energy at 298.15K | 2.50e-1 (± 2.02e-2) | 7.31e-4 (± 5.04e-4) | 3.44e-4 (± 2.18e-4) |
| GAT | Meta-learning | Rotational constant A | 6.65e-1 (± 9.57e-3) | 1.13e-3 (± 1.34e-3) | 1.37e-4 (± 1.64e-4) |
| GAT | Meta-learning | Rotational constant B | 3.24e-1 (± 4.79e-2) | 1.35e-2 (± 2.16e-2) | 8.79e-4 (± 2.83e-3) |
| GAT | Meta-learning | Rotational constant C | 3.36e-1 (± 3.02e-2) | 1.47e-2 (± 2.73e-2) | 6.78e-4 (± 9.86e-4) |
| MPNN | Meta-learning | Dipole moment | 3.82e-1 (± 2.10e-2) | 1.33e-3 (± 1.16e-3) | 2.98e-4 (± 2.18e-4) |
| MPNN | Meta-learning | Isotropic polarizability | 5.00e-1 (± 1.32e-2) | 1.32e-3 (± 1.10e-3) | 4.49e-4 (± 2.18e-4) |
| MPNN | Meta-learning | Highest occupied molecular orbital energy | 1.76e-2 (± 4.88e-3) | 4.26e-4 (± 3.32e-4) | 2.66e-4 (± 2.71e-4) |
| MPNN | Meta-learning | Lowest unoccupied molecular orbital energy | 6.56e-2 (± 9.28e-3) | 6.83e-4 (± 7.84e-4) | 4.78e-4 (± 6.16e-4) |
| MPNN | Meta-learning | Gap | 1.06e+0 (± 3.75e-2) | 1.78e-3 (± 1.44e-3) | 7.55e-4 (± 3.28e-4) |
| MPNN | Meta-learning | R2 | 4.22e-1 (± 3.37e-2) | 5.53e-3 (± 2.83e-3) | 3.95e-3 (± 2.53e-3) |
| MPNN | Meta-learning | Zero point vibrational energy | 4.13e-1 (± 2.29e-2) | 1.96e-3 (± 1.70e-3) | 5.87e-4 (± 5.24e-4) |
| MPNN | Meta-learning | Internal energy at 0K | 3.65e+0 (± 2.82e-1) | 3.11e-2 (± 1.84e-2) | 2.54e-2 (± 1.64e-2) |
| MPNN | Meta-learning | Internal energy at 298.15K | 5.99e+0 (± 3.58e-1) | 3.77e-2 (± 2.43e-2) | 2.81e-2 (± 2.07e-2) |
| MPNN | Meta-learning | Enthalpy at 298.15K | 3.24e+0 (± 2.75e-1) | 3.94e-2 (± 2.49e-2) | 2.43e-2 (± 1.95e-2) |
| MPNN | Meta-learning | Free energy at 298.15K | 4.95e+0 (± 3.00e-1) | 3.99e-2 (± 2.77e-2) | 2.79e-2 (± 2.57e-2) |
| MPNN | Meta-learning | Heat capacity at 298.15K | 6.85e-1 (± 2.05e-2) | 2.07e-3 (± 1.84e-3) | 5.80e-4 (± 3.21e-4) |
| MPNN | Meta-learning | Atomization energy at 0K | 7.23e-1 (± 1.88e-2) | 1.94e-3 (± 1.87e-3) | 4.79e-4 (± 3.16e-4) |
| MPNN | Meta-learning | Atomization energy at 298.15K | 2.51e-2 (± 3.19e-3) | 6.13e-4 (± 3.86e-4) | 5.28e-4 (± 3.61e-4) |
| MPNN | Meta-learning | Atomization enthalpy at 298.15K | 2.32e-1 (± 2.34e-2) | 8.32e-4 (± 5.18e-4) | 4.03e-4 (± 2.92e-4) |
| MPNN | Meta-learning | Atomization free energy at 298.15K | 1.35e+0 (± 4.58e-2) | 4.12e-3 (± 3.64e-2) | 1.43e-3 (± 8.61e-4) |
| MPNN | Meta-learning | Rotational constant A | 5.88e-1 (± 3.91e-2) | 1.96e-3 (± 1.70e-3) | 4.13e-4 (± 2.06e-4) |
| MPNN | Meta-learning | Rotational constant B | 1.65e-1 (± 1.82e-2) | 9.54e-4 (± 5.73e-4) | 5.49e-4 (± 2.67e-4) |
| MPNN | Meta-learning | Rotational constant C | 7.08e-2 (± 5.82e-3) | 4.69e-4 (± 2.52e-4) | 2.62e-4 (± 1.38e-4) |

the *Dipole moment*. In the table, we record how fast the model adapts to the new task (the loss with respect to the ground truth value) it has never seen as a function of the number of gradient updates used to optimize the model. Therefore, note that we are quickly learning entirely new tasks and at the same time, generalizing to a held-out set of molecules.

All models were training for 15,000 epochs. This was chosen as an arbitrary large number to guarantee convergence of the meta-learning algorithm. In practice, we observe 5,000 epochs to be enough. Indeed, past this number of training epochs performance plateaus. Experimentally we do not find any major difference in performance: performance on the train set does not substantially improve, and we do not see overfitting either.

Lastly, the Z-score normalization is computed by calculating the mean value for all the regression task labels as well as the standard deviation. Then all labels are normalized subtracting the calculated mean, and dividing by the standard deviation. Retrospectively, we acknowledge this may result in slight indirect information leakage given that quantities were computed across all tasks.

## D    Equivariant Message Passing Ensembles

Given the recent success of GNN architectures that exploit equivariance and invariance, such as [48] and [49], we also include some additional experiments using ensembles of equivariant MPNN models. We exploit the 3D coordinate information for each graph in the QM9 dataset. Using Equivariant MPNNs [47] we ensure layerwise equivariance to rotation and translations in 3D coordinates while preserving an overall invariant neural network. This architecture provides a beneficial strong inductive bias for our dataset. This is of special interest for datasets such as QM9 containing dynamical systems in which node coordinates are continuously being updated due to the action of intramolecular forces. This network uses three equivariant message passing layers, MLPs to model several non-linearities, and a global max pool aggregator at the end of the network.

### D.1    Details on Equivariant Message Passing Graph Neural Networks

We could naively attach the 3D coordinate information to the node features, but this would simply introduce noise; instead, one superior option is to implement layers that are invariant to 3D symmetry, such that

$$\mathbf{F}(\mathbf{H}, \mathbf{X}, \mathbf{A}) = \mathbf{F}(\mathbf{H}, \mathbf{X}\mathbf{Q} + \mathbf{T}, \mathbf{A}) \tag{1}$$

where $\mathbf{X}$ is a matrix of node coordinates for a given graph, $\mathbf{H}$ is the matrix of node features, $\mathbf{Q} \in \mathbb{R}^{3 \times 3}$ is an orthogonal rotation matrix, $\mathbf{T} \in \mathbb{R}^{3 \times 3}$ is a matrix with all its rows being equal to a translation vector $\mathbf{t} \in \mathbb{R}^3$, and $\mathbf{F}$ is a permutation equivariant function, following notation from [38, 39].

Note, however, applying layerwise equivariance to rotations and translations is even more effective [47], so that the following is satisfied

$$\mathbf{H}^{l+1}, \mathbf{X}^{l+1} = \mathbf{F}(\mathbf{H}^l, \mathbf{X}^l, \mathbf{A}) \rightarrow \mathbf{H}^{l+1}, \mathbf{X}^{l+1}\mathbf{Q} + \mathbf{T} = \mathbf{F}(\mathbf{H}^l, \mathbf{X}^l\mathbf{Q} + \mathbf{T}, \mathbf{A}). \tag{2}$$

A series of intricate updates are then computed by the equivariant message passing layer; details on these computations can be found in the treatise of [47], if interested.

### D.2    Results using Equivariant Message Passing Ensembles

We experiment with ensembles of meta-trained Equivariant MPNNs [47], where the number of models we aggregate ranges from 2 to 6. Table 7 displays the results. Note that in line with Table 3 from Section 4.3, the results are only testing on the *Dipole moment*. The ensembles of Equivariant MPNNs outperform those obtained using MPNNs in Section 4.3. For example, using learnable aggregation and combining 4 models, gives a loss of 1.66e-5 $\pm$ 1.22e-6 using Equivariant MPNNs. On the other hand, using ensembles of MPNNs we obtain a loss of 8.04e-5 $\pm$ 4.42e-5 after 5 gradient updates. This is expected since the Equivariant MPNNs can also leverage 3D coordinate information.

**Table 7:** Ensemble performance on QM9 dataset [44, 45] using Reptile [3] and Equivariant MPNNs. Values given are MSE $\pm$ standard deviation.

| No. Models ($M$) | Agg Method | Pre-Update | 1 Gradient Step | 5 Gradient Steps |
|---|---|---|---|---|
| 1 | N/A | 3.43e-1 ($\pm$ 1.12e-3) | 4.10e-4 ($\pm$ 4.70e-5) | 7.92e-5 ($\pm$ 3.81e-6) |
| 2 | Average | 2.67e-3 ($\pm$ 2.67e-4) | 7.44e-4 ($\pm$ 0.67e-4) | 2.08e-5 ($\pm$ 1.05e-6) |
| 2 | Learned | 2.67e-3 ($\pm$ 2.67e-4) | 7.08e-4 ($\pm$ 0.66e-4) | 1.95e-5 ($\pm$ 1.27e-6) |
| 4 | Average | 2.46e-3 ($\pm$ 2.99e-4) | 4.17e-4 ($\pm$ 1.72e-4) | 2.21e-5 ($\pm$ 1.32e-6) |
| 4 | Learned | 2.46e-3 ($\pm$ 2.99e-4) | 3.69e-4 ($\pm$ 1.33e-4) | 1.66e-5 ($\pm$ 1.22e-6) |
| 6 | Average | 2.20e-3 ($\pm$ 3.40e-4) | 2.08e-3 ($\pm$ 2.35e-4) | 2.41e-5 ($\pm$ 0.51e-5) |
| 6 | Learned | 2.20e-3 ($\pm$ 2.82e-4) | 2.01e-4 ($\pm$ 1.89e-5) | 1.09e-5 ($\pm$ 1.21e-6) |

