# OpenReview forum: "Graph Neural Network Expressivity and Meta-Learning for Molecular Property Regression"
_logconference.io/LOG/2022/Conference — LoG 2022 Poster_

### Official Review · Reviewer_7Pzf · 2022-10-17

**Overall Score:** 8
**Confidence:** 4

**Review:**

Reviews:

This paper extends the application of model-agnostic meta-learning (MAML) algorithms to graph neural networks (GNNs). In particular, it evaluates the performance of on few-shot graph-level regression tasks, specifically the molecular property prediction on Alchemy and QM9 datasets. The main contribution of this paper is the application of meta-learning to molecular modeling with comprehensive experiments.

Reasons for score:

Overall, I vote for accepting. I like the idea of making more effort on meta-learning a wide-range of molecule properties using GNNs, which can be potentially significant for drug discovery. My major concerns are that the motivation of this paper as well as some important additional experiments can be further complemented (see cons below). Hopefully the authors can address my concern in the rebuttal period.

Pros:

- This paper takes a step towards a very interesting while important topic: the meta-learning for few-shot molecular property prediction. For me, the problem itself is real and practical for both drug discovery and molecular modeling.
- Overall, the paper is well-structured. The authors state the problem clearly and make the experiments properly detailed.
- The experiments conducted in this work are comprehensive, which includes the Alchemy and QM9 dataset, with three GNNs being benchmarked and compared.

Cons:

- Previous works [1,2] of meta-learning for molecular property tasks focus on TOX21, SIDER, MUV, ToxCast datasets. The computational experiments conducted in this work can also be extended to these datasets for benchmarking, which make the conclusion about GNN expressivity robust and convincing.
- Any detailed illustrations and discussions after the empirical results are suggested to be appended. Since the work can belong to a practical topic in drug discovery, more explanation and analysis of the results can reveal important insights to the field.
- In fact, the proposed method lacks enough novelty. The MAML algorithm already stands as a general framework for arbitrary learnable model, includes the GNNs. Therefore this work can only be viewed as an application of MAML. On the other hand, there are already previous work focusing on meta-learning for molecular property prediction tasks, such as Meta-MGNN[1], PAR[2] and IterRefLSTM[3]. The authors are encouraged to make some meta-learning algorithm improvement that tailored for molecular modeling to make this work better motivated.
- Recently, the ML field have witnessed the fast development of the equivariant / invariant GNN architectures, GemNet[4], EGNN[5] to name a few. These models are very powerful to model the 3D geometric data and have achieved SOTA performances for property prediction tasks such as QM9 and OC20 datasets[6]. The authors are specially encouraged to benchmark and then discuss on how these GNNs that tailored for geometric graph (i.e., molecules) modeling behave under the MAML framework, which can be very interesting and remain untouched.

References:

- [1] Guo, Zhichun, et al. "Few-shot graph learning for molecular property prediction." *Proceedings of the Web Conference 2021*. 2021.
- [2] Wang, Yaqing, et al. "Property-aware relation networks for few-shot molecular property prediction." *Advances in Neural Information Processing Systems* 34 (2021): 17441-17454.
- [3] Han Altae-Tran, Bharath Ramsundar, Aneesh S Pappu, and Vijay Pande. Low data drug
discovery with one-shot learning. ACS Central Science, 3(4):283–293, 2017.
- [4] Gasteiger, Johannes, Florian Becker, and Stephan Günnemann. "Gemnet: Universal directional graph neural networks for molecules." *Advances in Neural Information Processing Systems* 34 (2021): 6790-6802.
- [5] Satorras, Vıctor Garcia, Emiel Hoogeboom, and Max Welling. "E (n) equivariant graph neural networks." *International conference on machine learning*. PMLR, 2021.
- [6] Chanussot, Lowik, et al. "Open catalyst 2020 (OC20) dataset and community challenges." *ACS Catalysis* 11.10 (2021): 6059-6072.

---

### Official Review · Reviewer_T9G7 · 2022-10-17

**Overall Score:** 6
**Confidence:** 3

**Review:**

Summary:

This work investigates meta learning for molecular property predictions with GNNs. In particular, the authors adopt Reptile on quantum chemistry (QM) property predictions with GCN, GAT, and MPNN. Experiments show that meta-learning significantly improves the performance of the GNNs under a few-shot learning setting. Also, the authors claim the performance of meta-learning is correlated with the expressivity of the GNN models (i.e., MPNN > GAT > GCN).

Strength:
- The problem that the authors probe is well-motivated. In molecular science, labeled data can be far from sufficient since experiments/simulations can be time-consuming. Meta learning can be an efficient way of make use of limited molecular data.
- This work investigates QM datasets (i.e., QM9 & Alchemy) which contain important properties besides previous classification tasks.
- This work adopts three different GNN models, which validates the effectiveness of meta learning on molecular property predictions.
- The manuscript is clearly written.

Issues:
- Tables 1&2 demonstrate the effectiveness of meta learning on few-shot learning setting. However, does meta learning initialized GNNs also converge to better results? Do the authors have results or comments on this?
- What is the input node and edge feature for the GNNs? Does 3D positional information included in the experiments? If not, how will 3D information affect the results?
- For equivariant GNNs that are designed for QM predictions using 3D molecular data, does meta learning also work? Do the authors have any comments on this?
- In line 69-75, the authors claim message-passing > attentional > convolutional in terms of GNN expressivity. I would recommend the authors formalize the definition of expressivity and provide references.
- This work is built upon existing meta learning method (Reptile) and GNN models (GCN, GAT, and MPNN). Though it is tested on QM datasets that have not been comprehensively investigated before, the technical contributions are still limited.

---

### Official Review · Reviewer_pKnW · 2022-10-21

**Overall Score:** 6
**Confidence:** 3

**Review:**

##########################################################################

Summary:

This paper applies the Reptile meta-learning framework to molecular property regression prediction tasks on two well-known benchmarking datasets. Specifically, it examines the role of GNN expressivity and ensembles in the outcome of this meta-learning process.

##########################################################################

Reasons for score (6):

This work presents novel and useful results based on appropriate experiments and offers reasoned analysis and explanations of the findings. I would rate this as a clear accept if the authors can clarify my concern about the train-validation-test splits.

##########################################################################

Strengths:

1. Good discussion of prior work to put this paper in context and justify why it is helpful/necessary. The authors provide good justification for using Reptile over MAML meta-learning (less computational overhead and more stable) and present results on regression tasks, while most prior work focused on binary classification tasks.

2. Good explanation of the different types of GNN (convolutional, attentional, and message passing), why some are more expressive than others, and what implications this is expected to have for meta learning.

3. The inclusion of error bars from the standard deviation on all results allows for drawing stronger conclusions about the statistical significance of differences between values.

4. The ensemble results are very promising, as they show a substantial reduction in error even for an ensemble size as small as 2 (with either learned or simple average weights).

##########################################################################

Weaknesses / major questions:

1. I did not see a discussion of train-validation-test splits here. Are all of the reported results based on training set performance? I'm not entirely familiar with the meta-learning literature, so I'm not sure what the standard practice is in this are of machine learning since the primary goal is to quickly learn entirely new tasks rather than to generalize to a held-out set on the same task. I think some discussion of this topic (i.e. whether any of the models presented might be overfit, and whether the meta-learned models are expected to generalize to unseen chemistries) is probably warranted. If train-validation-test splits were indeed used here, it would be helpful to provide more details on how those splits were done (e.g. random, scaffold, etc) and how the Z-score normalization was done to avoid leakage of training information into the test sets.

##########################################################################

Minor questions / suggestions:

1. (Line 49) I'm confused by the statement that second-order gradients are incorporated since the given equations seem to only include first-order gradients. Am I misunderstanding something here?

2. (Line 120) How was 15,000 epochs chosen? This seems like an extremely high number; how is overfitting prevented (related to weakness #1 above)?

3. (Table 1) Why was heat capacity at 298.15 K excluded from these results? The discussion in Appendix A (line 289), it seems that the primary reason is that it doesn't benefit much from meta-learning. Is that sufficient justification to exclude it? Is there any explanation of why if doesn't benefit as much?

#########################################################################

Typos:

* (Line 8) "emsembles" -> "ensembles"

#########################################################################

---

### Official Review · Reviewer_anaJ · 2022-10-22

**Overall Score:** 8
**Confidence:** 4

**Review:**

Accept

---

### Meta-Review · Area_Chair_Fq37 · 2022-11-15

**Confidence:** 4
**Recommendation:** Accept

**Meta Review:**

This paper presents a meta-learning framework for molecular property prediction. During discussion stage, authors have sufficiently addressed reviewers' concerns. The reviewers unanimously vote for acceptance and therefore the area chair decides to accept this paper.

---

### Decision · Program_Chairs · 2022-11-23

Accept (Poster)